# Properties Evaluations of Topology Optimized Functionally Graded Lattice Structures Fabricated by Selective Laser Melting

**DOI:** 10.3390/ma16041700

**Published:** 2023-02-17

**Authors:** Yangli Xu, Guangyao Han, Guoqin Huang, Tingting Li, Jiaxu Xia, Donghai Guo

**Affiliations:** 1Institute of Manufacturing Engineering, Huaqiao University, Xiamen 361021, China; 2Xiamen Institute of Software Technology, Xiamen 361024, China; 3Eplus 3D Tech (Beijing) Co., Ltd., Beijing 102206, China

**Keywords:** topology optimization, functionally graded lattice structures, selective laser melting, 316L stainless steel

## Abstract

Owning to their lightweight characteristic and high performance, functionally graded lattice structures (FGLSs) show great potential in orthopedics, automotive industries and aerospace applications. Here, two types of uniform lattice structures (ULSs) with RD = 0.50 and 0.20, and two types of FGLSs with RD = 0.30–0.50 and RD = 0.20–0.40, were designed by topology optimization and fabricated by SLM technology. Subsequently, their surface morphology, compressive deformation behavior and energy absorption abilities were evaluated by use of the finite element method (FEM) and compression tests. From these results, both elastic modulus and yield strength of specimens decreased with the lowering of the RD value. ULSs had a uniform deformation behavior with bending and bulking of struts, while FGLSs presented a mixed deformation behavior of different layers. Additionally, the energy absorption capability (*W_v_*) of specimens was proportional to the RD value. When the value of RD increased from 0.20 to 0.50, the *W_v_* of specimens increased from 0.3657 to 1.7469 MJ/m^3^. Furthermore, mathematical models were established successfully to predict the mechanical properties of FGLSs with percentage deviations < 10%. This work provides a comprehensive understanding regarding how to design and manufacture FGLSs with the properties desired for satisfying the demand of different application scenarios.

## 1. Introduction

Metallic functionally graded lattice structures (FGLSs) have caught researchers’ eyes due to their outstanding properties, such as high porosity (or low relative density, RD) for lightweight and liquid permeation, excellent ability to resist deformation, adequate strength for load-bearing requirements and high energy absorption for cushioning functions [1,2,3,4,5]. By varying their distributions of RD by means of cellular structures, FGLSs were recognized for the capability of achieving the desired mechanical properties for orthopedics, automotive industries and aerospace applications [6,7,8]. Generally, FGLSs are made using complex geometry, with interconnected rod-shape and surface-shape struts, so that such structures present difficulties when machined by conventional techniques. Additive manufacturing technology has been certified that can produce high-performance components by adjusting process parameters and adopting appropriated heat treatment [9]. In particular, selective laser melting (SLM) refers to processes that fabricate the parts through the addition of material in a layer-by layer technique, which has successfully fabricated complex lattice structures [10,11,12] and is considered an effective method to fabricate FGLSs.

The mechanical properties and energy absorption abilities of SLM-fabricated porous structures are greatly affected by several factors, such as porosity (or relative density), cellular structural type, and size. Under practical conditions (e.g., a simply supported beam or bone implants), the stresses in different areas are usually dissimilar. Therefore, in ensuring the designed structure meets the strength requirements while saving as much materials as possible, the relative density is designed differently in various regions. To date, most research interests involving this FGLSs using SLM fabrication have been concentrated on the effects of cellular structural types and metallic types on mechanical properties [13,14,15,16,17]. Mingkang Zhang et al. [18] designed and fabricated body-center-cubic (BCC) FGLSs based on Ti6Al4V alloy. They found that FGLSs with RD = 0.298 showed an elastic modulus ranging from 1.52 ± 0.1 GPa to 1.86 ± 0.23 GPa after heat treatment, and meanwhile achieved a high energy absorption of 76.40 ± 12.65 MJ/m^3^. Dheyaa S.J. Al-Saedi et al. [19] reported FGLSs with AlSi12 alloy exhibiting distinct deformation behavior and having higher energy absorption ability as compared to the uniform lattice structures. Triply periodic minimal surface (TPMS) was the most popular cellular structure due to its excellent properties and has prompted considerable interest for further investigation. Xiangyu Zhang et al. [20] investigated the mechanical properties and permeability of Ti6Al4V FGLSs. The results show that the permeability of FGLSs was mainly determined by the apparent RD, while the mechanical properties were affected by the loading mode. FGLSs exhibited superior deformability in the direction perpendicular to the compressive loading axle. Cong Zhang et al. [21] compared compressive behavior in uniform and graded lattice structures with 316L stainless steel. The results showed that during yield plastic deformation, the stress change in the graded structure is larger than that in the uniform structure.

Another important research focus of FGLSs is finding the mathematical relationship between the RD distributions of cellular structures and the mechanical properties of the whole lattice structure. One relevant mathematical method is the Gibson-Ashby model [22], which has been applied successfully to predict the mechanical properties of uniform lattice structures. However, the Gibson-Ashby model has proved to be inapplicable when predicting the mechanical properties of FGLSs. Recently, researchers tried to find an effective way to predict the mechanical properties of FGLSs. Lei Yang et al. [23] established a mathematical model to predict the elastic modulus and compressive strength of FGLSs with a density gradient perpendicular to the loading direction based on the Kelvin-Voigt model. However, this model cannot accurately predict the mechanical properties of FGLSs with a density gradient along the loading direction. A similar conclusion was reported by Xu and Zhong et al. [24,25]. Moreover, once the difference between the first layer and last layer of RD value expands to 0.1, the deviation of predicted and experimental properties would surpass 10% [26]. Therefore, establishing the appropriate model to predict the mechanical properties of FGLSs is a significant task.

Additionally, as mentioned above, many investigations of the properties of FGLSs based on different cellular structure types (BCC and TPMS) and materials (Ti6Al4V, Al alloy and 316L stainless steel) have been reported. However, very few papers have investigated the mechanical properties of FGLSs based on topology-optimized cellular structure. Topology Optimization (TO) is a structural optimization method that can establish a 3D model by controlling material distribution according to a defined special load condition and objective relative density [27]. Topology-optimized lattice structures have been reported to be structures with high elastic properties and controllable mechanical properties [28,29]. Until now, the compressive properties, deformation mechanism, absorption performance and properties prediction method of FGLSs using 316L stainless steel alloy have still not been investigated systematically.

This project seeks to identify the optimal means of evaluating and tailoring the compressive properties of topology-optimized 316L stainless steel FGLSs. To achieve this, two types of uniform lattice structures (ULSs) with RD = 0.50 and 0.20, as well as two types of functionally graded lattice structures (FGLSs) with RD = 0.30–0.50 and RD = 0.20–0.40, were fabricated by SLM technology and 316L stainless steel powder. Subsequently, their surface morphology, compressive deformation behavior, mechanical properties (elastic modulus and yield strength) and energy absorption abilities were evaluated using the finite element method (FEM) and compression tests. Moreover, the Gibson-Ashby model was applied to establish mechanical properties prediction model of ULSs. Based on this model, mechanical properties of ULSs with RD = 0.20–0.50 were calculated, and ultimately were used, to predict the mechanical properties of FGLSs. This work is helpful in designing the desired properties of FGLSs according to the demands of different application scenarios.

## 2. Materials and Methods

### 2.1. The Modelling of Lattice Structures

The detailed topology optimization process from one cellular structure to a whole lattice structure is shown in Figure 1:
①Topology optimization process. Consistent with our previous work [27,29], the cellular structures here were 3D-modeled by the optimization of a cube based on the topology optimization module of ABAQUS software. Material parameters modules were set corresponding to the mechanical properties of 316L stainless steel: the elastic modulus was 187 GPa and Poisson’s ratio was 0.33. The step module was set as the procedure type of “Static/General”. The vertex element of cube at the top right corner was subjected to the load (300 MPa, as this value cannot exceed the yield strength of 316L stainless steel, which is approximately 529 MPa), while its diagonal opposite vertex element was fixed completely. The RD of the objective model were constrained to 0.5, 0.4, 0.3, 0.2, and 0.10 of the original cube. After finishing this process, the 3D models of optimized cellular structures can therefore be output as .STL documents for further SLM fabrication.②Smooth process. All lattice structures, using the .STL file format, were input into Magics software to be repaired and smoothed.③Unit lattice structure modelling. The main method involved stacking 8 cellular structures in mirroring alignment with each other.④Lattice structure modelling. According to ISO 13314 standard [30], the lattice structures were established by repeating unit lattice structures with dimensions of 20 mm × 20 mm × 22 mm to produce specimens composed of 6 × 6 × 6 unit lattice structures, using Magics software.

### 2.2. SLM Fabrication of Designed Models

The commercial 316L stainless steel powder provided by SLM Solutions GmbH was used to fabricate specimens, which have a median particle size of 45 μm and a distribution from 15 μm to 63 μm. All designed lattice structures were manufactured by SLM 125^HL^ equipment from SLM Solutions GmbH (Lubeck, Germany). The build platform was machined from 316L stainless steel with a size of 125 mm × 125 mm × 25 mm. During SLM fabrication, the build volume was always kept at an oxygen level < 100 ppm by purging with argon. The SLM fabrication parameters of specimens were set as: laser power of 195 W, preheating temperature of 200 °C, layer thickness of 30 μm, scanning speed of 1100 mm/s, hatch distance of 0.12 mm, and scanning direction rotated by 67° alternately between the layers. The SLM processing parameters were selected based on information from the original equipment manufacturer, and are considered to be the optimal process parameters for additive manufacturing 316L stainless steel alloy. The whole SLM fabrication process was monitored by the Melt Pool Monitoring System (MPMS). MPMS is an on-axis tool for visualizing the thermal radiation distribution of the melt pool in the SLM process, and able to evaluate the quality of the production process. When the specimens were completely manufactured, specimens were removed from build platform by wire electrical discharge machining. Finally, to ensure that residual powder particles in the internal region of specimens could be removed completely, sandblasting and ultrasonic cleaning using ethyl alcohol were used.

Two types of lattice structures were designed; as shown in Figure 2, they are ULSs with RD = 0.50 {U(0.50)} and 0.20 {U(0.20)}, as well as FGLSs with RD = 0.30–0.50 {G(0.30–0.50)} and RD = 0.20–0.40 {G(0.20–0.40)}. Three independent specimens were SLM-fabricated for each set of structural parameters and only one group of them are shown here. Additionally, in order to determine the manufacturing and designing limit of RD, a lattice structure with RD = 0.15–0.50 was also fabricated.

### 2.3. Stress Distribution Simulations

To investigate the deformation mechanism of lattice structures in the compression test, the finite element method (FEM) was applied using ABAQUS 2020 software to simulate the stress distribution on the surface of lattice structure models under unidirectional compression. Additionally, it should be noted that the experimental stress-stain curve and mechanical properties of dense 316L stainless steel were shown in Figure 3 which were obtained from Lei Zhang’s work [31]. The 3D structural solid element of the 4-node tetrahedral type was used to mesh the models with six degrees of freedom per node. In these simulation models, the direction of compression was set as “down along the *Z*-axis”. Subsequently, a rigid plate was placed on the bottom of the models to incorporate boundary conditions, and the interaction between the rigid plates with the models were defined as tie constrained. Boundary conditions were set as: the bottom plate of the model was restricted in all degree of freedom, while the nodes in other regions were allowed to move in the compression process. The FEM simulations of lattice structures were limited to the elastic period, so only 0.2 mm displacement along the *Z*-axis was designed within the models.

### 2.4. Measurements of Relative Density (RD)

The value of RD corresponding to different specimens was measured by Densitometer MH-3100 (China) and calculated via Archimedes’ principle [32]. The average value of RD was obtained by performing three calculations for each specimen, using Equation (1):(1)$$\mathrm{RD}=\frac{M_{a}\times \rho_{water}}{\left(M_{a}-M_{b}\right)\times \rho_{316L}}$$

In Equation (1), *M_a_* is the mass of the specimen in air (g); *M_b_* is the mass of SLM-fabricated specimen in water; *ρ_water_* is the density of water (1 g/cm^3^); and *ρ_316L_* is the density of dense 316L stainless steel (7.98 g/cm^3^). By measuring the mass of the specimens in water and air, and then substituting those values into Equation (1), the value of RD can be obtained.

### 2.5. Surface Morphology Measurements and Compression Test

The surface morphology of specimens was observed by using a HITACHI SU1510 scanning electron microscope (SEM) under 30 kV from HITACHI company (Tokyo, Japan). According to the standard ISO 13314, a compression test was carried out by a ProLine-Z100 electronic universal testing machine at room temperature from ZWICK ROELL company (Ulm, Germany). Stress-strain curves, mechanical properties and energy absorption properties of specimens were plotted and calculated by corresponding data obtained from the compression test. In order to obtain detailed deformation images during the compressive process, the whole compression test was recorded by a digital video.

## 3. Results and Discussion

### 3.1. Fabrication Results of Specimens

The entirety of the SLM fabrication process of specimens was monitored by MPMS, and thermal radiation distributions of melt pool of lattice structures for different layers are shown in Figure 4. Essential to note is that the color’s change in the MPMS results represents temperature intensity distribution in one layer during the SLM process; the measurement data recorded by MPMS is not the real temperature of the melt pool. The red color represents “hot spots”, and the blue color is “cold spots”. Hot spots derive from the excessive accumulation of energy deposition, which can lead to generated thermal stress and deform the parts. Cold spots can arise from shading of the laser due to the smoke, which leads to a lower energy deposition and a dimmer monitoring signal. The exterior areas of cold spots show that the balling effect or the incomplete lapping of melt pool occurs in SLM process. The hot spots and cold spots detected by MPMS can be clearly associated with the fabrication quality of the real parts. Therefore, the optimal fabrication quality is indicated when the value of the temperature intensity of SLM-fabricated specimens is constantly between that of the hot spots and that of the cold spots. In this work, the temperature intensities of all the specimens were kept in a range from 800 to 1400, and the whole SLM process was therefore smooth and without the balling effect and deformation of specimens. This indicates that the setting of SLM fabrication parameters was reasonable.

Figure 5 shows fabrication results and SEM images of lattice structure with RD = 0.15–0.50, namely, G(0.15–0.50). Based on experimental results, the cellular structures with RDs from 0.20 to 0.50 can be fabricated completely. Because the heat affected the zone of laser beam, it can be seen in Figure 5b that some residual powder particles adhered to the surface of the specimens. This is a typical phenomenon with SLM technology; metallic powder particles would be partially melted and bonded to the surface of specimens due to the heat effect from the contour of the laser beam spot [9,10]. In addition, this phenomenon leads to the measured RD values of specimens being slightly larger than those of the designed models (shown in Figure 6). However, the cellular structures with RD = 0.15 were fabricated defectively, that is, the rod-shaped struts were not connected completely. This is because the diameter of broken rod-shape struts is approximately 300 μm, which may cause problems during the SLM process.

### 3.2. Compressive Properties of Specimens

#### 3.2.1. Stress-Strain Curves

Figure 7 displays the stress-strain curves of ULSs and FGLSs with various RD values as determined by compressive tests. Typically, three deformation regions of porous materials can be seen: the linear elastic stage, the yield plastic deformation stage, and the densification stage. According to the Gibson-Ashby theory [24], the compressive stress-strain curves for metallic porous materials may obey the law of elastic-plastic foam, which has a yield plastic deformation stage. Specially, the stress-strain curves of all specimens here presented a yield plastic deformation stage with a certain dip angle, which was defined as the law of elastomeric foam (e.g., foam rubber). For topology-optimized cellular structures, similar results were reported by Zefeng Xiao et al. [28]. From their experimental results, only SLM-fabricated 316L stainless steel lattice structures with RD > 0.25 had stress-strain curves of elastomeric foam, while those with RD < 0.25 obeyed elastic-plastic stress-strain curves. On the other hand, for Gyroid cellular structures by TPMS design, the stress-strain curves of specimens with RD = 0.15 evinced the law of elastic-plastic foam. This illustrated that topology-optimized structures generally have superior performance. Specifically, in this work, even specimens with RD = 0.20 still present elastomeric stress-strain curves. This indicates that the designed structures have strong resistance to deformation.

As expected, the stress-strain curves of FGLSs were positioned between those of ULSs with RD = 0.5 and 0.20. The stress-strain curves of specimens seem to be relative to their RD. As the RD decreases, four main differences arise: Firstly, the slope of a straight line fitted to the elastic stage, which is defined as the elastic modulus of materials, gradually becomes smaller and smaller. This indicates that the ability to resist the deformation weakened as RD declined. Secondly, the length of the yield plastic deformation stage became longer with the decrease in RD. Thirdly, the densification of specimens with lower RD values obviously appeared later. These two experimental results arose from the important fact that the specimens with lower RD had higher porosity, which needs greater strain to compress the density. Finally, the “α”, which was defined as the angle between the yield plastic deformation stage and the axis of strain, became smaller with the lessening of RD. Similar results were reported by Zefeng Xiao et al. [26], also indicating a weaker load-bearing ability.

Figure 8a,b shows the calculated value of elastic modulus (*E*) and yield strength (*σ_y_*) based on stress-strain curves of specimens, respectively. The yield strength here was calculated by finding the intersection of the stress-strain curve and a line parallel to the linear quasi-elastic curve at a strain offset of 0.2% [30]. Obviously, either *E* or *σ_y_* of specimens decreased with decreased RD value. For ULSs, U(0.50) possessed the mechanical properties (*E* = 5.042 ± 0.050 GPa, *σ_y_* = 120.241 ± 0.283 MPa), and U(0.20) possessed the mechanical properties (*E* = 0.74 ± 0.013 GPa, *σ_y_* = 29.433 ± 0.802 MPa). For FGLSs, G(0.30–0.50) possessed the mechanical properties (*E* = 3.148 ± 0.055 GPa, *σ_y_* = 76.924 ± 0.401 MPa), G(0.30–0.50) possessed the mechanical properties (*E* = 1.609 ± 0.020 GPa, *σ_y_* = 37.820 ± 0.705 MPa). Interestingly, for ULSs filled with the same cellular structures and using the same based-material (316L stainless steel), both *E* and *σ_y_* of the specimens in this work were approximately two times higher than those in our previous work, except for U(0.20) [29]. This is because the number of cellular structures filling into ULSs of this work was 10 × 10 × 10, while the number of cellular structures filling into ULSs of reference [29] was 6 × 6 × 10. The number of cellular structures is the main factor in explaining the significant difference in mechanical properties of ULSs.

#### 3.2.2. Deformation Behavior

The deformation behavior of lattice structures is mainly affected by material type, relative density and cellular structure type. Generally, the effect of relative density on deformation behavior is considered in terms of the size or thickness modifications of struts in lattice structures [13]. Moreover, lattice structures with the RD gradient distribution should present a different deformation behavior compared with those with the RD uniform distribution, which would be rendered in detail, as discussed below.

Figure 9 displays the camera images of deformation process of ULSs with RD = 0.20 and RD = 0.50, as well as FGLSs with RD = 0.30–0.50 and RD = 0.20–0.40. Additionally, the stress distribution FEM results of ULSs and FGLSs at 2% overall strain under compressive testing are shown in Figure 10. From these results, the deformation images matched well with the stress-strain curves of specimens shown in Figure 7. Obviously, ULSs presented a different deformation behavior compared with FGLSs. For U(0.50), some thin struts began to bend, which resulted in a slight buckling in the linear elastic stage. Subsequently, in the yield plastic deformation stage, a certain number of thin struts were broken, and thick struts began to be the main load-bearing actor. This viewpoint is also supported by FEM results. From Figure 10a, it can be seen that severe stress is mainly concentrated in the regions corresponding to thin struts, while relatively lower stress is found in regions corresponding to the thick struts. Therefore, structural crack nucleation and growth would be more likely to appear in thin struts and finally, result in fracture. Additionally, as thick struts can bear larger compression and are more difficult to break, the line of the stress-strain curves presented a yield plastic deformation stage with a certain dip angle (as shown in Figure 7). Upon further compressing U(0.50), the final densification stage corresponding to rapidly increasing stress was eventually reached. The same deformation behavior for U(0.20) can be seen in Figure 9b and Figure 10b. In distinction, the bulking degree of U(0.20) seems to be more severe than that of U(0.50), which may be due to the weaker struts and higher porosity of U(0.20).

Specifically, FGLSs specimens displayed a significant difference in deformation behavior compared with ULSs. For G(0.30–0.50), which consisted of cellular structures with RD = 0.30–0.50, presenting a mixed deformation behavior for different layers. As shown in Figure 9c, some thin struts started to bend, leading to a little buckling during the linear elastic stage. When the compressive process of specimens entered into the yield plastic deformation stage, the layers with high RD were gradually compressed to a high density, whereas those with low RD were still bending and bulking. Finally, the layers with high RD entered the densification stage earlier than did those with low RD. This phenomenon can be explained by FEM results, as shown in Figure 10c. At the beginning of the compression test, the region filled with cellular structures with high RD values experienced higher stress concentration than those areas having low RD values. Additionally, it is a common viewpoint that specimens with higher RD have lower porosity and are more easily compressed. This is also in agreement with previous experimental results (Figure 7). Similar deformation behavior of G(0.20–0.40) can be observed from Figure 9d and Figure 10d. The main difference in deformation behavior between G(0.20–0.40) and G(0.30–0.50) is that when compressive process entered into the densification stage, the cellular structures with low RD for G(0.20–0.40) were largely broken and hinged together, while the cellular structures for G(0.30–0.50) were mainly compressed from bucking at high density, albeit without fractured struts. This indicates that the deformation behavior might be relative to the size of the struts. Similar conclusions were reported by Tianlin Tong et al. [25] and Lei yang et al. [33]. Tianlin Tong et al. proposed that the struts on the specimens which have much smaller diameters are more susceptible to collapse. Meanwhile, based on simulated and experimental results, Lei Yang et al. investigated the fracture of porous structures and found that cracking was most likely caused by the tensile stresses that were concentrated within the region of smaller struts.

For further investigation of the deformation mechanism of FGLSs, the surface morphologies of struts of G(0.20–0.40) were observed after the compression test by SEM equipment, as shown in Figure 11. Figure 10b shows the SEM observation of cellular structures with high RD, whereas Figure 10c,d display fracture surfaces of cellular structures with low RD. From these SEM results, previous experimental conclusions were again verified, specifically, that layers with high RD presented a trend of being compressed to high density while those with low RD tend to be broken. Moreover, typical ductile fracture surfaces with very fine dimple sizes (<1 μm) can be seen in Figure 11d. This indicates that thinner struts with low RD have better ductility and are easier broken than are thicker struts with a higher RD value.

### 3.3. Energy Absorption Properties of Specimens

The investigation of the energy absorption capability of ULSs and FGLSs plays an important part in evaluating the impact resistance and stability of designed structures. The cumulative value (*W_v_*) of energy absorption is defined as:(2)$$W_{v}=\frac{1}{100}\overset{e_{0}}{\underset{0}{\int }}\sigma \mathrm{de}$$
where *W_v_* is the cumulative energy absorption per unit volume (MJ/m^3^), σ is the compressive stress (MPa), $e_{0}$ is upper limit of the compressive strain, $\sigma_{0}$ is the compressive stress at the upper limit of the compressive strain (N/mm^2^), and the $e_{0}$ here represents the strain of 0.5 during compression test.

Figure 12 presents the energy absorption per unit volume-strain curves of different specimens calculated by stress-strain curves (Figure 7). As expected, the energy absorption abilities of designed structures were affected by their RD values, which were also reported by Xiaojie Fan et al. [14]. When the value of RD increased from 0.20 to 0.50, the *W_v_* of ULSs increased from 0.3657 to 1.7469 MJ/m^3^. A similar trend can be seen for FGLSs specimens. When the average value of RD increased from 0.287 to 0.385, the *W_v_* of FGLSs changed from 0.6372 to 1.1801 MJ/m^3^. This can be explained by the fact that lattice structures with higher RD value have thicker struts which can bear larger stress, so the height of the yield plateau stage in stress-strain curves becomes larger, and shows higher energy absorption abilities compared with specimens of low RD value.

### 3.4. Properties Prediction Models of FGLSs

#### 3.4.1. Properties Prediction Models of ULSs

It is important to establish the mechanical properties-densities relationship of designed structures for technological applications. For ULSs, the Gibson-Ashby model was utilized to establish the relationship between relative mechanical properties {elastic modulus (E) and yield strength ($\sigma_{y}$)} and RD ($\rho /\rho_{s}$), which obey the following functions:(3)$$\frac{E}{E_{s}}=C_{1}{\left(\frac{\rho }{\rho_{s}}\right)}^{n_{E}}$$
(4)$$\frac{\sigma_{y}}{\sigma_{s}}=C_{2}{\left(\frac{\rho }{\rho_{s}}\right)}^{n_{\sigma }}$$
where $E_{s}$, $\sigma_{s}$ and $\rho_{s}$ represent the elastic modulus, compressive strength and density of bulk 316L stainless steel, respectively. Here, the values of $E_{s}$ and $\sigma_{s}$ are defined as 187 GPa and 529 MPa based on tensile testing [29]. *C*_1_ and *C*_2_ need to be calculated, and are related to the geometric shape of porous materials. Additionally, for better calculating efficiency, $n_{E}$ was set as the value of 2, whereas $n_{\sigma }$ was set as the value of 1.5 [23]. In this work, the mechanical properties of U(0.20) were applied to calculate the Gibson-Ashby models, and the properties of U(0.50) were used to verify them.

The fitted curves describing the relationship between relative densities and mechanical properties of ULSs specimens are plotted from the data of Figure 8, as shown in Figure 13 and Figure 14. The Gibson-Ashby models of ULSs were determined when the value of *C*_1_ = 0.10 and *C*_2_ = 0.63 were calculated according to the properties of U(0.20). Subsequently, the other predicted values of mechanical properties of ULSs with RD from 0.20 to 0.50 were calculated based on these models, as listed in Table 1. Specifically, the mechanical properties of U(0.50) were plotted in Figure 13 and Figure 14, showing a moderately higher value in elastic modulus but a values closely aligned with predicted ones. This indicates that the prediction models are in good agreement with experimental results. For further evaluating the properties of FGLSs, their experimental mechanical properties were also plotted in Figure 13 and Figure 14. Interestingly, the relative elastic modulus of both G(0.20–0.40) and G(0.30–0.50) have obviously higher values than the values predicted for the ULSs. This result indicates that grade density distribution perpendicular to the direction of compression would improve the elastic properties of porous lattice structures. The same conclusions were reported by Lei Yang et al. [23]. However, the relative yield strengths of G(0.20–0.40) are a little lower than the predicted value of ULSs. These results provide an important reference for designing high-performance structures. Lattice structures with RD grade distribution are a recommended design that can obtain higher elastic modulus aside from the plastic deformation stage.

#### 3.4.2. Properties Prediction Models of FGLSs

The high-performance mechanical properties of FGLSs present an attractive and favorable prospect in lightweight design fields. Therefore, it is important to find a way to predict the mechanical properties of FGLSs. As mentioned previously, the Gibson-Ashby model is not practicable for establishing the relationship between mechanical properties and the relative densities of FGLSs. Another classical theory, namely the Kelvin-Voigt model, can be used to predict the elastic modulus and strength of the FGLSs. Based on this model, the mechanical properties are determined as the weighted average of properties of all layers. The elastic modulus and yield strength of FGLSs can therefore be described as:(5)$$E_{G}=\overset{n}{\underset{1}{\sum }}k_{n}\times E_{n}$$
(6)$$\sigma_{G}=\overset{n}{\underset{1}{\sum }}k_{n}\times \sigma_{n}$$
where *n* represents the layer number, which here is equal to five layers, with each layer corresponding to one RD. For example, for G(0.20–0.40) as shown in Figure 2, the RD values of the first and second layer are 0.20, and the third and fourth layer are 0.30, while the fifth layer is 0.40. *k_n_* represents the volume percentage occupied by the layer with respect to the whole structure, which here is designed to be 20% for every layer (there are five layers in total). *E_n_* and *σ_n_* are the properties of ULSs corresponding to relative density of layers, as listed in Table 1.

Table 2 lists the experimental and predicted mechanical properties of FGLSs. The deviation here is defined as (average experimental value-predicted value)/(average experimental value)×100%. The predicted elastic modulus and yield strength of G(0.30–0.50) were 2.908 GPa and 79.805 MPa with percentage deviations of 7.62% and 3.75%, respectively. Separately, the predicted elastic modulus and yield strengths of G(0.20–0.40) were 1.593 GPa and 50.707 MPa, with percentage deviations of 0.99% and 34.07%, respectively. The experimental properties matched fairly well with the predicted value predicted by the Kelvin-Voigt models, except for the yield strength of G(0.20–0.40).

This phenomenon can be explained as follows. At the elastic stage, each layer with different RD bore the compressing together, and the transition of RD did not affect the structural integrity of the FGLSs, so their elastic modulus could be accurately predicted by Kelvin-Voigt model. However, once entering the yield plastic stage, FGLSs showed a different deformation behavior, in that layers with higher RD achieved the densification stage early and generated work by hardening before the failure of the whole structure. The layers with lower RD broke, which would decrease the strength of specimens. Therefore, the predicted values of yield strength were notably lower than the experimental values, such as G(0.20–0.40).

The prediction models of FGLSs in this work are not absolutely ideal, due to other factors that would affect the predicting accuracy, such as SLM process parameters, materials, the geometrical shape of designed cellular structure and RD distribution range. Therefore, more experimental tests should be carried out to validate and refine the prediction model for further applications.

## 4. Conclusions

Because of their lightweight nature and high performance, FGLSs will likely play an increasingly important role in various industrial fields. For their further application, it is critical to understand how to predict the mechanical properties of FGLSs. In this work, ULSs and FGLSs with different RD distributions were designed by topology optimization and fabricated by SLM technology. Their compressive behavior, energy absorption abilities and modeled properties prediction were investigated. The conclusions can be set forth as following.

(1)SLM-fabricated specimens were in good agreement as to geometric morphology with the designed models. However, only the cellular structures with RD from 0.20 to 0.50 could be fabricated completely. The cellular structures with RD = 0.15 were fabricated defectively because the diameter of broken rod-shape struts is approximately 300 μm, which might cause problems during the SLM fabrication process.(2)Both the elastic modulus and the yield strength of specimens decreased with the lessening of the RD value. For ULSs, U(0.50) possessed the mechanical properties (*E* = 5.042 ± 0.050 GPa, *σ_y_* = 120.241 ± 0.283 MPa), and U(0.20) possessed the mechanical properties (*E* = 0.74 ± 0.013 GPa, *σ_y_* = 29.433 ± 0.802 MPa). For FGLSs, G(0.30–0.50) possessed the mechanical properties (*E* = 3.148 ± 0.055 GPa, *σ_y_* = 76.924 ± 0.401 MPa), G(0.30–0.50) possessed the mechanical properties (*E* = 1.609 ± 0.020 GPa, *σ_y_* = 37.820 ± 0.705 MPa).(3)The stress-strain curves of all specimens followed that of the elastomeric foam mode, indicating that the designed structures had a strong resistance to deformation. ULSs had a uniform deformation behavior with bending and bulking of struts in different layers, while FGLSs specimens displayed a significant difference in deformation mechanism compared with ULSs, presenting a mixed deformation behavior in different layers.(4)The energy absorption capability (*W_v_*) of specimens was proportional to their RD value. When the value of RD increased from 0.20 to 0.50, the *W_v_* of ULSs increased from 0.3657 to 1.7469 MJ/m^3^. When the average value of RD increased from 0.287 to 0.385, the *W_v_* of FGLSs changed from 0.6372 to 1.1801 MJ/m^3^. This can be explained by the fact that lattice structures with higher RD value have thicker struts, which can bear larger stress.(5)The Gibson-Ashby model was introduced to precisely estimate the mechanical properties of ULSs, with values of *C_1_* = 0.10 and *C_2_* = 0.63. The Kelvin-Voigt model was proved to efficiently have predicted the elastic modulus and yield strength of FGLSs, with percentage deviations < 10%.

## Figures and Tables

**Figure 1 materials-16-01700-f001:**
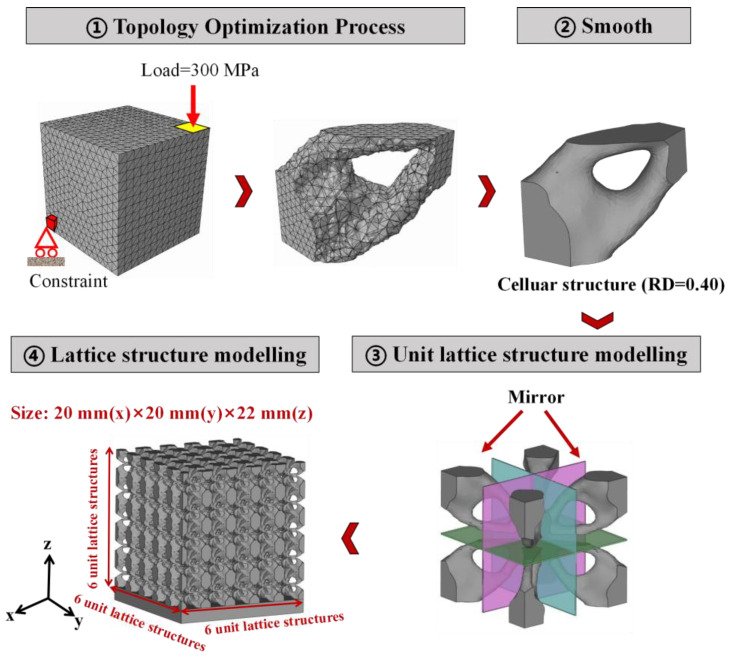
Detailed topology optimization process of lattice structures.

**Figure 2 materials-16-01700-f002:**
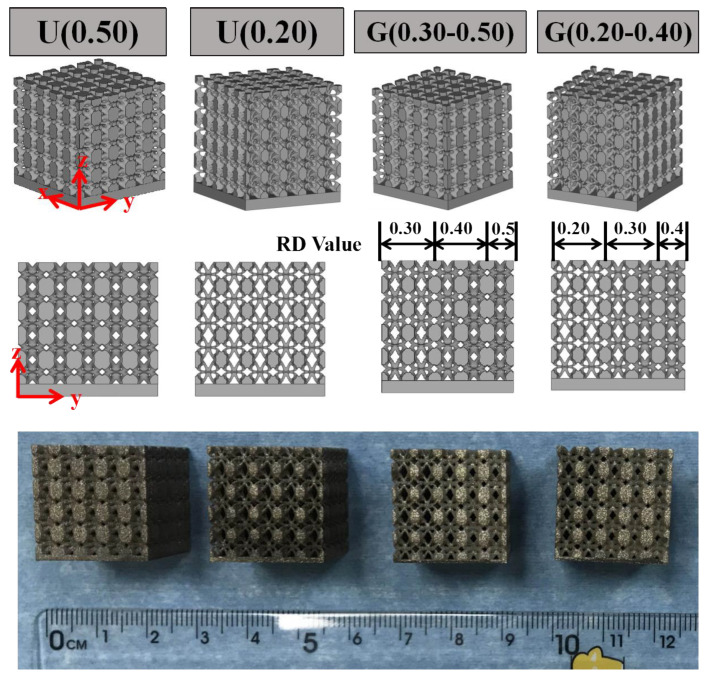
The CAD models and SLM fabrication of designed lattice structures.

**Figure 3 materials-16-01700-f003:**
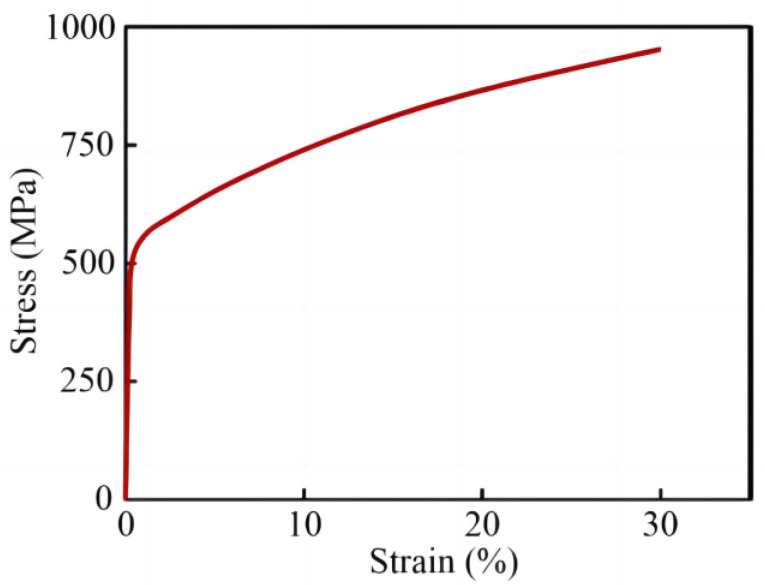
The experimental compressive stress-strain curve of SLM-fabricated dense 316L stainless steel [31].

**Figure 4 materials-16-01700-f004:**
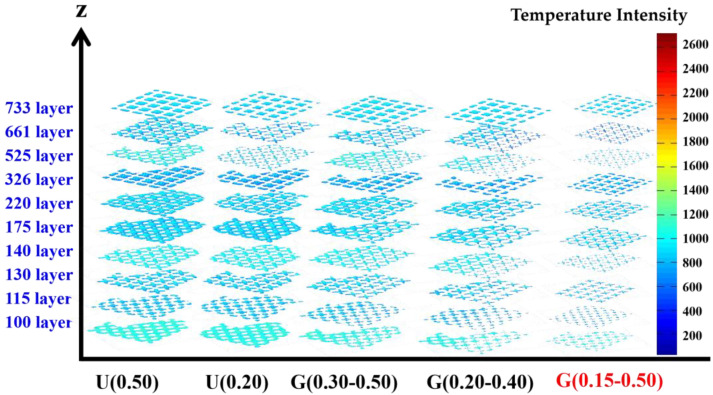
Temperature intensity distribution of lattice structures monitored by MPM module.

**Figure 5 materials-16-01700-f005:**
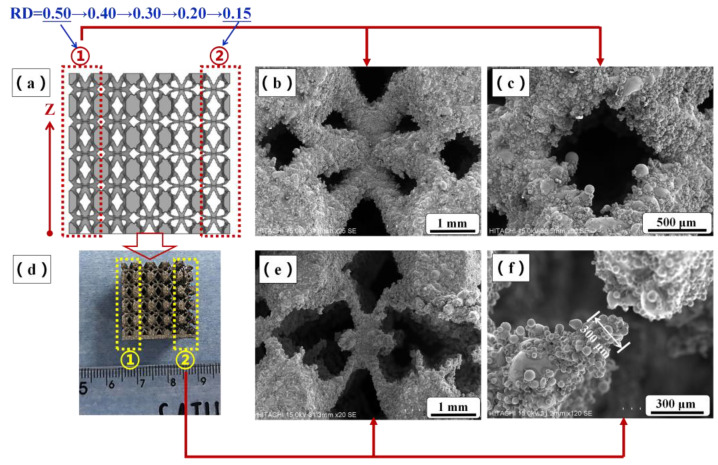
(**a**) CAD model of test lattice structure with RD from 0.15–0.50, (**b**,**c**) SEM micrographs of lattice structure in region ①, (**d**) SLM fabrication result of test lattice structure, and (**e**,**f**) SEM micrographs of lattice structure in region ②.

**Figure 6 materials-16-01700-f006:**
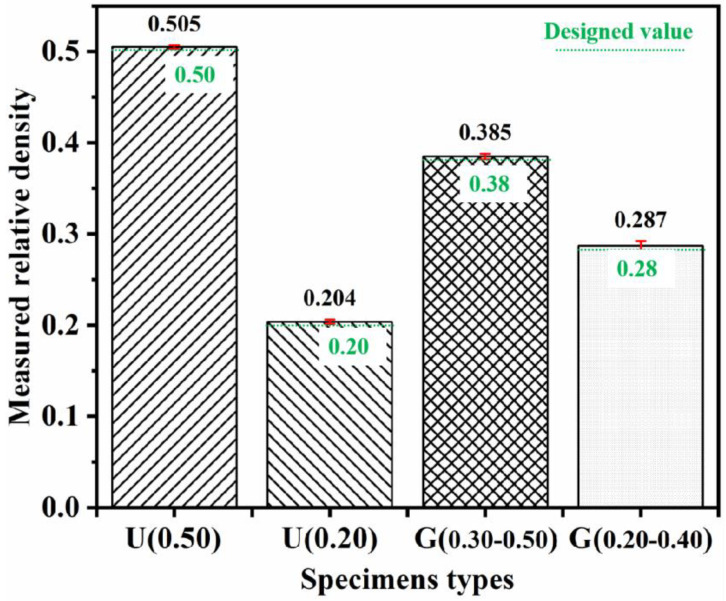
The measured (black) and designed (green) relative densities of lattice structures.

**Figure 7 materials-16-01700-f007:**
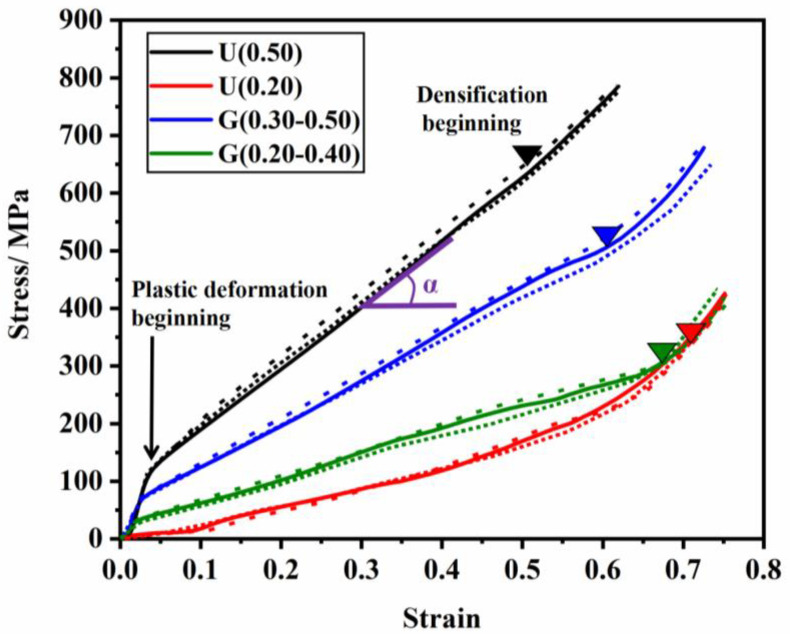
The stress-strain curves of lattice structures by compression test.

**Figure 8 materials-16-01700-f008:**
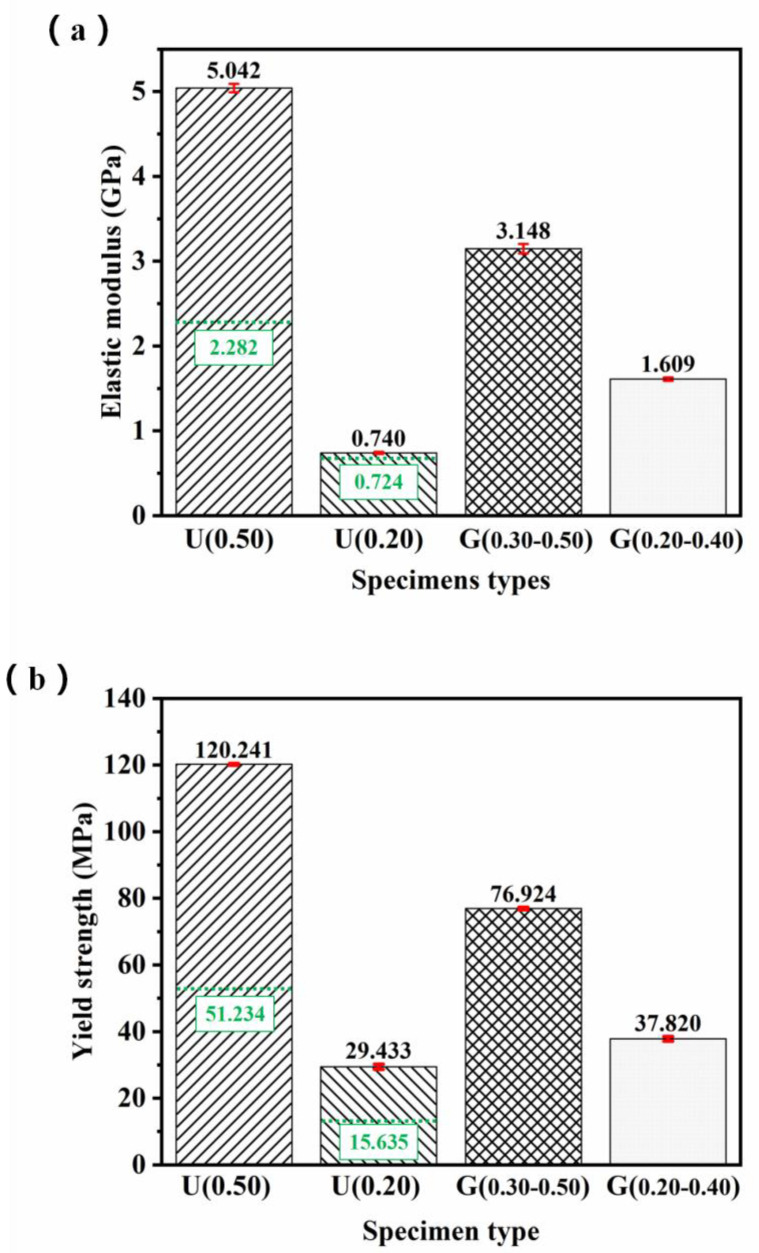
(**a**) The elastic modulus and (**b**) yield strength of lattice structures. The green datas are from reference [27].

**Figure 9 materials-16-01700-f009:**
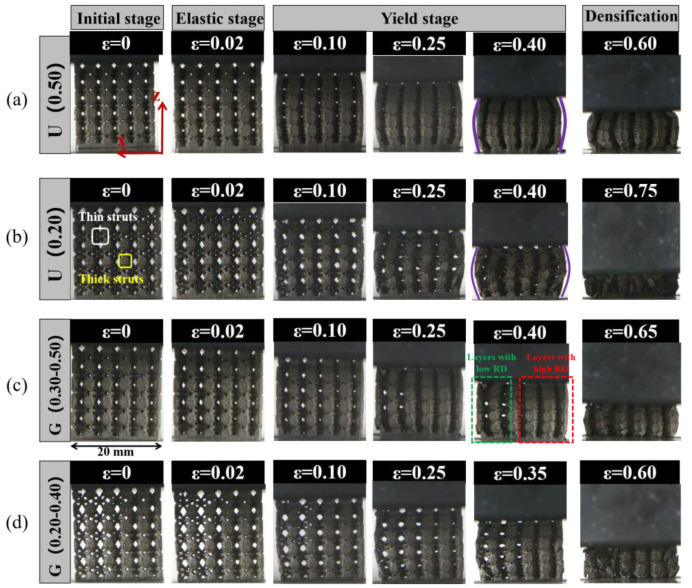
Compressive deformation behaviors images of different lattice structures under increasing strain: (**a**) U(0.50), (**b**) U(0.20), (**c**) G(0.30–0.50), (**d**) G(0.20–0.40).

**Figure 10 materials-16-01700-f010:**
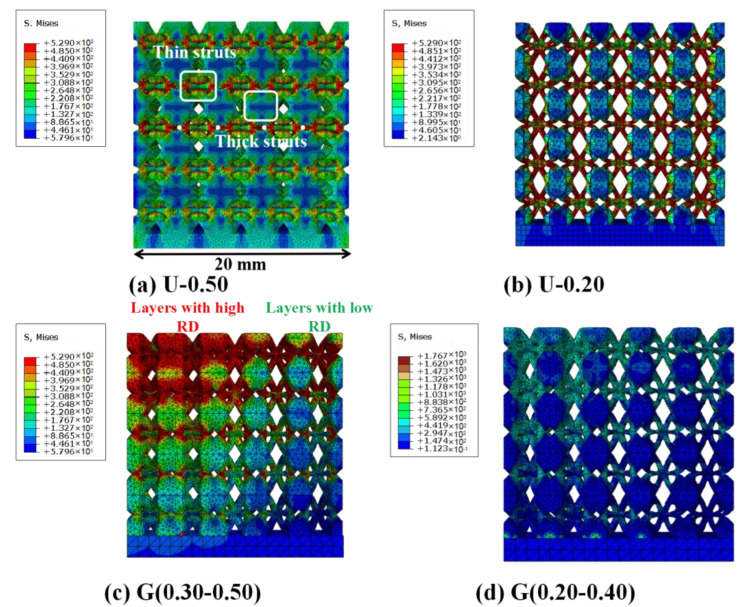
FEM results of maximum principal stress distribution on the surface of lattice structures at 0.02 strain under compressive loading: (**a**) U(0.50), (**b**) U(0.20), (**c**) G(0.30–0.50) and (**d**) G(0.20–0.40).

**Figure 11 materials-16-01700-f011:**
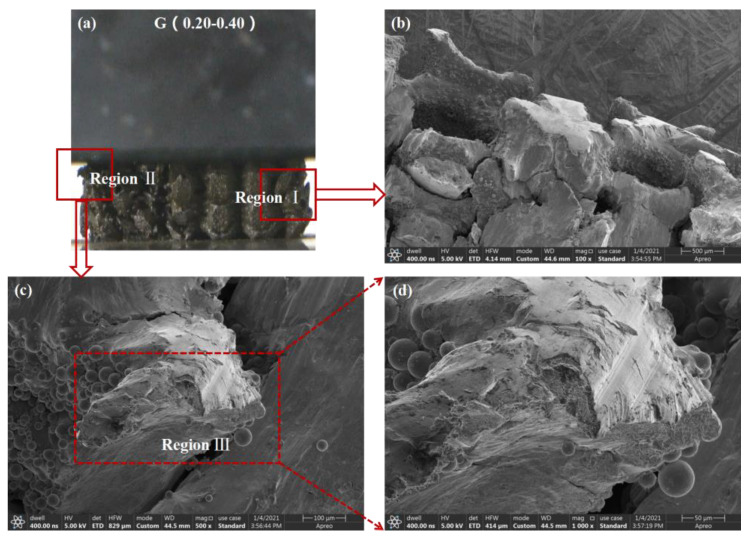
SEM surface observations of G(0.20–0.40) in different regions after compression test. (**a**) the compressive process image of G(0.20–0.40) in final densification stage, (**b**) the SEM image of region Ⅰ, (**c**) the SEM image of region Ⅱ (**d**) the SEM image of region Ⅲ.

**Figure 12 materials-16-01700-f012:**
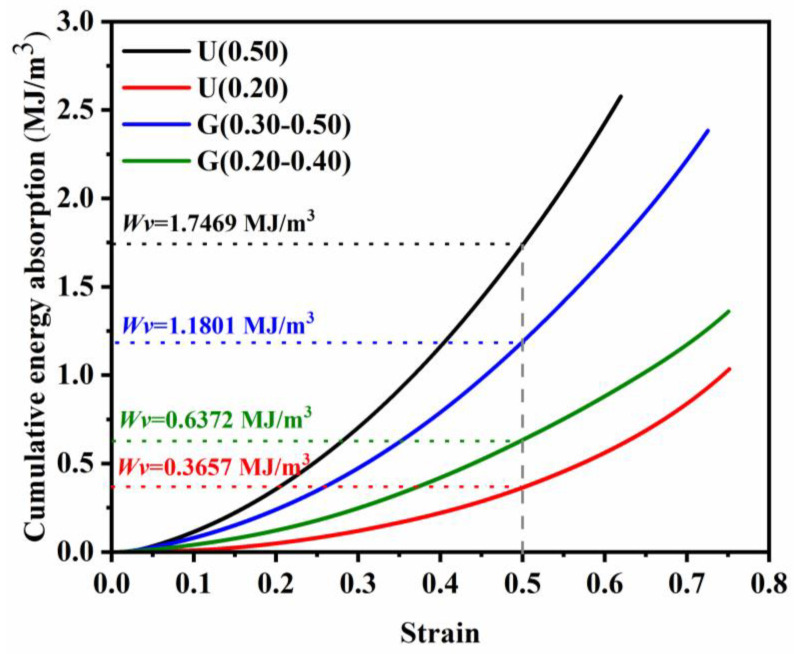
Cumulative energy absorption for different lattice structures.

**Figure 13 materials-16-01700-f013:**
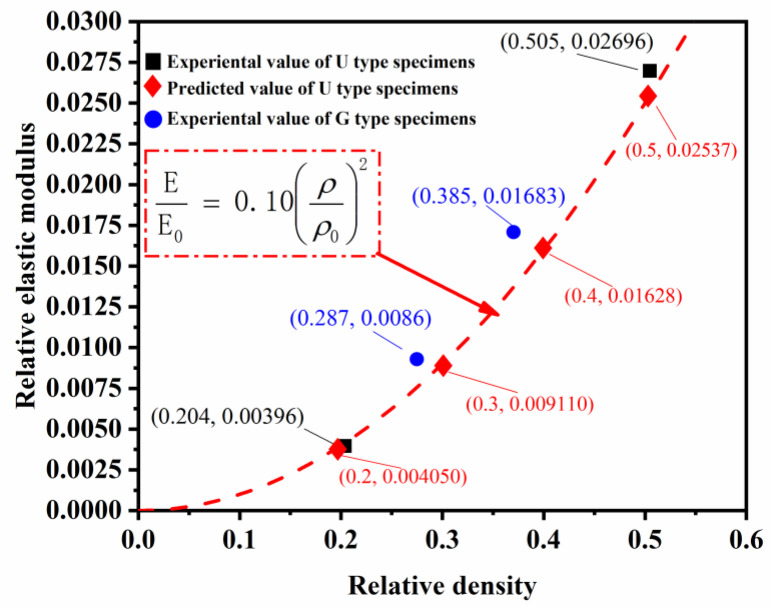
Relative elastic modulus-relative density fitted curves of ULSs.

**Figure 14 materials-16-01700-f014:**
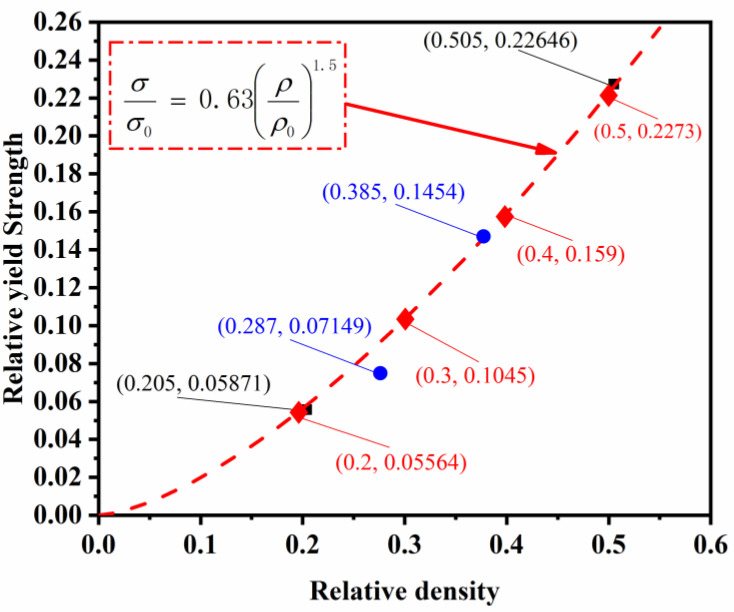
Relative yield-strength-relative density fitted curves of ULSs.

**Table 1 materials-16-01700-t001:** Mechanical properties of ULSs with different RD values, as predicted from Gibson-Ashby model.

RD value	0.50	0.40	0.30	0.20
Elastic modulus, GPa	5.042	3.044	1.704	0.757
Yield strength, MPa	120.242	84.111	55.281	29.434

**Table 2 materials-16-01700-t002:** Comparisons of experimental and predicted mechanical properties of FGLSs.

Property	G(0.30–0.50)			G(0.20–0.40)		
	Experimental	Predicted	Deviation	Experimental	Predicted	Deviation
*E* (GPa)	3.148	2.908	7.62%	1.609	1.593	0.99%
*σ_y_* (MPa)	76.924	79.805	3.75%	37.820	50.707	34.07%

## Data Availability

Not applicable.

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
