# Peer review of "Properties Evaluations of Topology Optimized Functionally Graded Lattice Structures Fabricated by Selective Laser Melting"

_materials, 2023, doi:10.3390/ma16041700_

Round 1
Reviewer 1 Report
The abstract should be written more attractively. Most of it contains research methods. Almost all of the abstract is devoted to generalities and research methods. The novelty of the article should be clearly added to the abstract. Use quantitative results in the abstract.
What is the purpose of the design? Do the results indicate the achievement of the desired goal?
Compared to other sections, the introduction has been written in summary, which can be more comprehensive. Use the following sources. “The high temperature flow behavior of additively manufactured Inconel 625 superalloy” --- “Review of Selective Laser Melting of Magnesium Alloys: Advantages, Microstructure and Mechanical Characterizations, Defects, Challenges, and Applications”)
How is the reproducibility of mechanical properties result? How many samples have been tested?
The conclusion section should be modified like the abstract.
In the results analysis section, can the current research results be compared with previous models?
What is the basis for selecting printing parameters?
Discuss more the failure and fracture modes of samples.
Is the macroscopic photo of the samples prepared during the compression test? It is suggested to use these images.
Add a scale bar to some figures such as SEM images (Figure 10).
Author Response
Dear Editors and Reviewers,
Thank you for your comments concerning our manuscript (materials-2126694). Those comments are all valuable and very helpful for revising and improving our paper, as well as the important guiding significance to our work. We have studied comments carefully and made revisions. Revised parts have been marked using red color in the revised manuscript. The main revisions and the responds to your comments are as following:
Comment 1: The abstract should be written more attractively. Most of it contains research methods. Almost all of the abstract is devoted to generalities and research methods. The novelty of the article should be clearly added to the abstract. Use quantitative results in the abstract.
Response:Thank you for your valuable comments. We feel very sorry that we ignored the important information such as quantitative results or the novelty in the abstract. Now, we have revised the abstract carefully. However, the requirement of “the abstract” from materials journal is : a single paragraph of about 200 words maximum. I hope you can accept it that some contents of this work can not be presented clearly. Thank you very much!
Comment 2: What is the purpose of the design? Do the results indicate the achievement of the desired goal?
Response: Thank you for your careful reading of our work. The purpose of this work is to find out a method to predict the mechanical properties of Functionally Graded Lattice Structures (FGLSs) designed by topology optimization. From our previous work (Xu, Y.; Li, T.; Cao, X.; Tan, Y.; Luo, P. Compressive Properties of 316L Stainless Steel Topology Optimized Lattice Structures Fabricated by Selective Laser Melting. Adv. Eng. Mater. 2020, 23, 2000957.), topology optimized lattice structures have excellent mechanical properties and energy absorption abilities compared with some other structures. Also, the mechanical properties of Uniform Lattice Structures (ULSs) have been proved that can be predict successfully by Gibson-Ashby model. However, the designed structures is used to fabricate a multi-level lattice structure (or gradient lattice structure) for the demand of different application scenarios, for example, bone has a gradient structure which can be simply classified to cancellous bone and cortical bone. Therefore, this work focus on investigating the compressive behavior, energy abilities and properties prediction of Functionally Graded Lattice Structures. For predicting mechanical properties of FGLSs, Gibson-Ashby model was firstly used to establish mechanical properties prediction model of ULSs. Based on this model, mechanical properties of ULSs with RD = 0.20-0.50 were calculated. By Kelvin-Voigt model, corresponding mathematical models were established successfully for predicting mechanical properties of FGLSs, with percentage deviations < 10%. So, this work is significant because it provides a efficient method to design FGLSs which can tailor mechanical properties according to the different load-bearing demand.
Comment 3: Compared to other sections, the introduction has been written in summary, which can be more comprehensive. Use the following sources. “The high temperature flow behavior of additively manufactured Inconel 625 superalloy” --- “Review of Selective Laser Melting of Magnesium Alloys: Advantages, Microstructure and Mechanical Characterizations, Defects, Challenges, and Applications”)
Response:Thank you for your valuable comments. I have read these two papers carefully and rewritten “the introduction”. I really appreciate the opportunity to improve my work. Thank you again!
Comment 4: How is the reproducibility of mechanical properties result? How many samples have been tested?
Response: Thank you for your careful reading. In this work, three independent specimens of each structural parameters were SLM-fabricated for compression test. In Figure 7, three strain-stress curves of each type specimens can be seen and has relatively high consistency. Figure 8 a and b shows the calculated value of elastic modulus (E) and yield strength (σy) based on stress-strain curves of specimens, respectively. For ULSs, U(0.50) possessed the mechanical properties (E=5.042±0.050 GPa, σy=120.241±0.283 MPa), and U(0.20) possessed the mechanical properties (E=0.74±0.013 GPa, σy=29.433±0.802 MPa). For FGLSs, G(0.30-0.50) possessed the mechanical properties (E=3.148±0.055 GPa, σy=76.924±0.401 MPa), G(0.30-0.50) possessed the mechanical properties (E=1.609±0.020 GPa, σy=37.820±0.705 MPa). This results indicated the mechanical properties result has the reproducibility. Thank you very much!
Comment 5: The conclusion section should be modified like the abstract.
Response:Thank you for your valuable comments. Now, we have revised the conclusion section carefully. I hope you can accept it if some contents of this work were not presented clearly. Thank you very much!
Comment 6: In the results analysis section, can the current research results be compared with previous models?
Response: Thank you for your careful reading of our manuscript. Just as you said, it is very important to compare the difference of current research designed models and previous models. In fact, we have reported the mechanical properties difference between designed lattice structures in this work and other method designed models, such as FCC, VC, BCCZ, Groid and so on(Xu, Y.; Li, T.; Cao, X.; Tan, Y.; Luo, P. Compressive Properties of 316L Stainless Steel Topology Optimized Lattice Structures Fabricated by Selective Laser Melting. Adv. Eng. Mater. 2020, 23, 2000957.). The main conclusion is that compared with other lattice structures, the designed lattice structures have superior performance compared with most of the other structures because the unnecessary material in non-loading region were removed by topology optimization. This work is an extension of previous work. So, in this work, we concentrate on discuss the method to predict mechanical properties of FGLSs and did not compare the difference of current research designed models and previous models.
Comment 7: What is the basis for selecting printing parameters?
Response: Thank you for your careful reading. I feel terribly sorry we ignored the details of the SLM processing parameters selected. The SLM processing parameters was selected based on the original equipment manufacturer (SLM Solutions GmbH, Germany), which is considered to be the optimal process parameters for additive manufacturing 316L stainless steel alloy. Now, we have added some important details to describe this method in section 2.2.
Comment 8: Discuss more the failure and fracture modes of samples.
Response: Thank you for your instructive suggestions. We feel very sorry that we have not explained the failure and fracture modes of specimens clearly. According to my understanding, they have large relationship with the whole deformation behavior of lattice structures, as described in section 3.2.2. In this work, failure and fracture mechanisms of specimens need to be discussed separately. The failure modes of specimens were mainly affected by the RD distribution. For ULSs, which filled with cellular structures of uniform RD value, some thin struts began to bend and resulted in slightly buckling in linear elastic stage. Subsequently, in yield plastic deformation stage, a certain amount of thin struts were broken and thick struts began to be the main load-bearing actor. The final densification stage corresponding to rapidly increasing stress generated eventually. However, for FGLSs, which were consisted of cellular structures with RD = 0.30-0.50 or 0.20-0.40, presenting a mixed deformation behavior for different layers. some thin struts started to bend and lead to a little buckling during linear elastic stage. When compressive process of specimens entered into yield plastic deformation stage, the layers with high RD gradually were compressed to dense, whereas those with low RD still were bending and bulking. Finally, the layers with high RD were earlier to enter densification stage than those with low RD. Differently, the fracture mechanisms of specimens may have closely relative to the size of struts. The cellular structures with low RD having thin struts tend to occur largely broken and be hinged together, while the cellular structures having thick struts mainly were compressed from bucking to dense without struts fracture. These two fracture behavior lead to the densification stage of lattice structure, resulting in structural failure.
Comment 9: Is the macroscopic photo of the samples prepared during the compression test? It is suggested to use these images.
Response: Thank you for your instructive suggestions. We have add more macroscopic photo of samples during the compression test, as shown in Figure 9.
Comment 10: Add a scale bar to some figures such as SEM images (Figure 10).
Response: Thank you for your instructive suggestions. I have checked all figures and revised them.
All in all, the editors and reviewer’s comments are quite helpful. I look forward to hearing from you, and receiving more advisable comments. Thank you for your instructive advice again.

Reviewer 2 Report
The authors studied the use of selective laser melting for fabrication of functionally graded structures. While the manuscript is generally well executed, there are several issues that should be addressed before further consideration for publication.
1. Suggest the authors to use ISO/ASTM standard terminology when describing the additive manufacturing process.
2. In line 85, it should be "316L stainless steel powder"
3. What is the meaning of "smooth process"? Is it smoothening of the surfaces by reducing the triangulation in the STL file?
4. Line 112, should be "lattice"
5. Finite element analysis has been used for study of lattice structures fabricated by SLM. Suggest the authors to discuss if there is any new contribution from this portion of the work? How about the boundary conditions and parameters used for the FEM?
6. By using Equation 1, do you get the porosity within the struts of the lattices or the overall lattice porosity?
7. Any discussion on the deviations between the designed samples and actual fabricated samples? And any characterisation on this aspect? What about defects within the samples? How will they affect the results?
Author Response
Dear Editors and Reviewers,
Thank you for your comments concerning our manuscript (materials-2126694). Those comments are all valuable and very helpful for revising and improving our paper, as well as the important guiding significance to our work. We have studied comments carefully and made revisions. Revised parts have been marked using blue color in the revised manuscript. The main revisions and the responds to your comments are as following:
The authors studied the use of selective laser melting for fabrication of functionally graded structures. While the manuscript is generally well executed, there are several issues that should be addressed before further consideration for publication.
Comment 1: Suggest the authors to use ISO/ASTM standard terminology when describing the additive manufacturing process.
Response: Thank you for your valuable comments. I have read the ISO/ASTM 52921-2013<standard terminology for additive manufacturing - coordinate systems and test methodologies >, and check our work carefully. Now, I have revised some description of additive manufacturing process, which marked by blue word.
Comment 2: In line 85, it should be "316L stainless steel powder"
Response:Thank you for your valuable comments. We have check manuscript carefully and revised this mistake.
Comment 3: What is the meaning of "smooth process"? Is it smoothening of the surfaces by reducing the triangulation in the STL file?
Response: Thank you for your careful reading. The meaning of “smooth process” is surface repair of three-dimensional models. Surface repair was carried out on the three-dimensional model of STL sheet whose damage was not too serious. The holes on the surface of STL were filled, the edges on the surface of STL were removed, and the surface of the object in STL format was treated as smooth, and the rough model was transformed into a fine model. I hope it helps to answer your questions.
Comment 4: Line 112, should be "lattice"
Response:Thank you for your valuable comments. We have check manuscript carefully and revised this mistake.
Comment 5: Finite element analysis has been used for study of lattice structures fabricated by SLM. Suggest the authors to discuss if there is any new contribution from this portion of the work? How about the boundary conditions and parameters used for the FEM?
Response: Thank you for your careful reading. Finite element method was also made into a specialized modules in ABAQUS software. We just input the mechanical properties (modulus and strength), tensile stress-strain curve of 316L stainless steel alloy,meshing the lattice structures’ model (the 3D structural solid element of the 4-node tetrahedral type), setting boundary conditions ( the bottom plate of the model was restricted in all degree of freedom, while the nodes in other region were allowed to translate in compressive loading direction), and finally obtaining the stress distribution on the surface of specimens with a displacement of 0.2 mm. FEM method was applied in this work to analyze the stress distribution of specimens at 2% overall strain under compressive testing (Figure 10). The main purpose here is to explain the deformation trend and fracture sensitive regions of specimens would be generated. These FEM results were considered to be closed to experimental results and matched well with deformation process images in Figure 9. However, the FEM results can not simulate the entire compression process accurately. I hope you can accept it because this model somehow explains the compression process and allows us to improve it in the future work. Thank you very much!
Comment 6: By using Equation 1, do you get the porosity within the struts of the lattices or the overall lattice porosity?
Response: Thank you for your careful reading. By using Equation 1, we got the overall lattice porosity. The porosity within the struts of the lattices can not measured completely because the hole is inside the struts. And the relative density (numerically equal to <1- potosity> of lattice structure) in this work was calculated based on Archimedes’principle using water, the detailed approach was as follows:Archimedes' Principle, law of physics that states that when an object is totally or partially immersed in a fluid, it experiences an upthrust (F=Mbg, g=9.8 N/kg) equal to the weight of the fluid displaced. So, by measuring the weight of the fluid displaced, we can know the mass (Mb) of SLM-fabricated specimen in water. In addition, by measuring the mass (Ma) of the specimens in air, then substituting it into the Eq. (1), the value of relative density can be calculated. Eq.(1) is derived from the literature of 【Journal of Alloys and Compounds 782 (2019) 209-223】.
Comment 7: Any discussion on the deviations between the designed samples and actual fabricated samples? And any characterisation on this aspect? What about defects within the samples? How will they affect the results?
Response: Thank you for your careful reading. From our work, the measured RD values of specimens were slightly larger than those of designed models (Figure 6). And this is a common understanding that metallic powder particles would be melt partly and bonded to the surface of specimens due to the heat effect of out contour of laser beam spot for SLM technology. This is the main factor that lead the deviations between the designed samples and actual fabricated samples. We realize that this difference in RD will affect the mechanical properties evaluation of lattice structures, so the relative density deviations were taken into account when fitting the Gibson-Ashby model (Figure 13 and 14). It should be pointed out that the RD measuring method in this work can not measure the porosity inside the lattice structure. However, there are indeed a large number of small defects (such as micro holes) in the structure. These defects are the growth points of cracks, which will greatly reduce the service performance and life of parts. The usual solution is to optimize SLM process parameters or adopt heat treatment to repair defects. To author’s knowledge, the defects of parts using SLM fabrication is still unresolved well. All in all, it should be acknowledged that the defects of lattice structures were not described in detail here because the main purpose of this work is to investigate the compressive properties of lattice structures. I hope you can accept it and allows us to improve it in the future work. Thank you very much!
All in all, the editors and reviewer’s comments are quite helpful. I look forward to hearing from you, and receiving more advisable comments. Thank you for your instructive advice again.

Reviewer 3 Report
The paper is ready for publishing after some minor changes
Figure 2a is too small. I think that a better diagram is needed or as it something that it is common knowledge it can be removed.
Figure 3 Although it is stated that the values are from reference 29. I think that it would help the reader if a small explanation about the data (it is just experimental or it has some numerical modelling)
Section 2.4 Experimental setup and equipment used could be added. In the rest of the sections everything is well described, it is true that it is a simple experiment but that would help to understand the experimental uncertainty.
Figure 4 Should be bigger (with better quality, temperature scale has some strange gray background.
Figure 6 and corresponding text. It is stated that RD values are slightly larger than those of designed models, it is clear in Uniform model but expected density of gradient porosity models should be included.
Figure 10 and corresponding text. I would have liked to see FEM simulations with model deformation. Do you have them or those are just FEM static simulations? Please explain this point because it its the most relevant for me.
Table 1 is missing its caption. It has the one from the template.
I think that it is a good paper in which some of the methods are presented in a brief way (with references too previous works), but I find it really interesting.
The changes that I suggest can improve the quality of the paper but it could be published as it is .
Author Response
Dear Editors and Reviewers,
Thank you for your comments concerning our manuscript (materials-2126694). Those comments are all valuable and very helpful for revising and improving our paper, as well as the important guiding significance to our work. We have studied comments carefully and made revisions. Revised parts have been marked using purple color in the revised manuscript. The main revisions and the responds to your comments are as following:
The paper is ready for publishing after some minor changes
Comment 1: Figure 2a is too small. I think that a better diagram is needed or as it something that it is common knowledge it can be removed.
Response: Thank you for your instructive suggestions. I have removed Figure 2a.
Comment 2: Figure 3 Although it is stated that the values are from reference 29. I think that it would help the reader if a small explanation about the data (it is just experimental or it has some numerical modelling)
Response: Thank you for your instructive suggestions. The the values are from reference 29 is experimental stress-stain curve of dense 316L stainless steel. I have added explain in revised manuscript.
Comment 3: Section 2.4 Experimental setup and equipment used could be added. In the rest of the sections everything is well described, it is true that it is a simple experiment but that would help to understand the experimental uncertainty.
Response: Thank you for your instructive suggestions. The RD value of specimens were measured by Densitometer MH-3100 (China), which was added in revised manuscript.
Comment 4: Figure 4 Should be bigger (with better quality, temperature scale has some strange gray background.
Response:Thank you for your instructive suggestions. We have revised it now.
Comment 5: Figure 6 and corresponding text. It is stated that RD values are slightly larger than those of designed models, it is clear in Uniform model but expected density of gradient porosity models should be included.
Response: Thank you for your instructive suggestions. We have revised it now.
Comment 6: Figure 10 and corresponding text. I would have liked to see FEM simulations with model deformation. Do you have them or those are just FEM static simulations? Please explain this point because it its the most relevant for me.
Response: Thank you for your careful reading of our manuscript. I can not provide the FEM simulations with model deformation because the FEM method was applied in this work to analyze the stress distribution of specimens at 2% overall strain under compressive testing (Figure 10). Therefore, The FEM simulations of lattice structures were limited to the elastic period, so only 0.2 mm displacement along the Z-axis was set to the models. At this level, the model deformation is not obvious. The main purpose here is to explain the deformation trend and fracture sensitive regions of specimens by the stress distribution of specimens would be generated. I hope you can accept it and allows us to improve it in the future work. Thank you very much!
Comment 7: Table 1 is missing its caption. It has the one from the template.
Response: Thank you for your careful reading of our manuscript. We have corrected it now.
Comment 8: I think that it is a good paper in which some of the methods are presented in a brief way (with references too previous works), but I find it really interesting. The changes that I suggest can improve the quality of the paper but it could be published as it is.
Response: Thank you for your comments again.
All in all, the editors and reviewer’s comments are quite helpful. I look forward to hearing from you, and receiving more advisable comments. Thank you for your instructive advice again.

Round 2
Reviewer 1 Report
Some of the comments from the previous step were not well considered. (comments 3 and 8).
Comment 3: Compared to other sections, the introduction has been written in summary, which can be more comprehensive. Use the following sources. “The high temperature flow behavior of additively manufactured Inconel 625 superalloy” --- “Review of Selective Laser Melting of Magnesium Alloys: Advantages, Microstructure and Mechanical Characterizations, Defects, Challenges, and Applications”)
Comment 8: Discuss more the failure and fracture modes of samples.
Author Response
Reviewer #1:
Dear Reviewer,
Thank you for your comments concerning our manuscript (materials-2126694). Those comments are all valuable and very helpful for revising and improving our paper, as well as the important guiding significance to our work. We fell so sorry that Some of the comments (comments 3 and 8) from the previous step were not well considered. Now, we have studied comments carefully and made revisions. Revised parts have been marked using Green color in the revised manuscript. The main revisions and the responds to your comments are as following:
Comment 3: Compared to other sections, the introduction has been written in summary, which can be more comprehensive. Use the following sources. “The high temperature flow behavior of additively manufactured Inconel 625 superalloy” --- “Review of Selective Laser Melting of Magnesium Alloys: Advantages, Microstructure and Mechanical Characterizations, Defects, Challenges, and Applications”)
Response:Thank you for your valuable comments. I fell so sorry that we did not understand the meaning of your comment very well.
For the paper of “The high temperature flow behavior of additively manufactured Inconel 625 superalloy”, the authors deals with the high temperature flow behavior of an additive manufactured Inconel 625 superalloy in a wide temperature range of 800-1100 ℃ under of the various strain tates of 0.001-0.1 s-1. The strain hardening rate analysis was utilized to reveal the critical condition for the occurrence of the restoration processes. The additively manufactured material was capable to be dynamically recrystallized at various thermomechanical conditions. By investigating the effect of the process parameters on the high temperature flow behavior of an additively manufactured nickel base superalloy and obtaining a proper phenomenological constitutive based model to precise prediction of the flow stress levels. This paper shows that additive manufacturing technology can produce high-performance components by adjusting process parameters. We think it is very important and added the introduction.
For the paper of “Review of Selective Laser Melting of Magnesium Alloys: Advantages, Microstructure and Mechanical Characterizations, Defects, Challenges, and Applications”, authors proposed that Selective Laser Melting (SLM) is considered a more reliable way of producing Mg-based products, especially for applications with complex geometry design, the least amount of waste, and no need for molds and accessories. Meanwhile, this paper summarizes the SLM of Mg by introducing the SLM parameters, properties, defects, and applications of SLMed Mg alloys and discussing the challenges and solutions of this method. I think it also can help to explain the effect of thermal zones produced in the SLM process of alloy on the relative densities difference between designed lattice structures and SLM-fabricated specimens. Therefore, we cited this paper into our work as well.
All in all, we must admit that the introduction is not very well written. We are also trying to improve it. I hope you can temporarily accept its status quo. I really appreciate the opportunity to improve my work. Thank you again!
Comment 8: Discuss more the failure and fracture modes of samples.
Response: Thank you for your instructive suggestions. We feel very sorry that we have not well explained the failure and fracture modes of specimens clearly. One of the main finding of this work is that the failure modes of specimens were mainly affected by the RD distribution. I think I have explained it clearly in our previous respond. However, we may not explain well about the fracture mechanisms of specimens and how it is influenced by the size of struts. In this parts, we still hold this perspective that the fracture mechanisms of specimens is that cellular structures with low RD having thin struts tend to occur largely broken and be hinged together, while the cellular structures having thick struts mainly were compressed from bucking to dense without struts fracture. These two fracture behavior lead to the densification stage of lattice structure, resulting in structural failure. These conclusions were common understanding for explaining the fracture mechanism of 316 L stainless steel lattice structures. Tianlin Tong et al. proposed that the struts on the specimens which have much smaller in diameters are more susceptible to collapse[Zhong, T.; He, K.; Li, H.; Yang, L. Mechanical properties of lightweight 316L stainless steel lattice structures fabricated by selective laser melting. Materials & Design 2019, 181, 108076.]. Meanwhile, based on simulated and experimental results, Yang et al. reported the fracture of porous structures were found that cracking was most likely caused by the tensile stresses that were concentrated at the region of smaller struts [Yang, L.; Yan, C.; Cao, W.; Liu, Z.; Song, B.; Wen, S.; Zhang, C.; Shi, Y.; Yang, S. Compression compression fatigue behaviour of gyroid-type triply periodic minimal surface porous structures fabricated by selective laser melting. Acta Mater. 2019, 181, 49-66]. I think these two papers can support our viewpoint and explain the fracture mechanism of 316L stainless steel lattice structures fabricated by selective laser melting. Meanwhile, we have cited these two paper in our work. At the same time, I think we may not have explained it very well. I hope you can forgive our shortcomings. We will continue to improve our level in future work. Thank you again!
All in all, your comments are all quite helpful. Thank you for your instructive advice again.
Sincerely,
Yangli Xu
Address:
Hua Qiao University,
668 Jimei Avenue,
Jimei District,
Xiamen,
Fujian, 361021
P. R. China
E-mail: ylxu@hqu.edu.cn
Tel: +86 130 2102 9011

Reviewer 2 Report
NIL
Author Response
Thank you for your useful comments and suggestions! Best wishes to you!